# Localization and Single Molecule Dynamics of *Bacillus subtilis* Penicillin-Binding Proteins Depend on Substrate Availability and Are Affected by Stress Conditions

**DOI:** 10.3390/cells14060429

**Published:** 2025-03-13

**Authors:** Lisa Stuckenschneider, Peter L. Graumann

**Affiliations:** 1SYNMIKRO, Zentrum für Synthetische Mikrobiologie, Hans-Meerwein-Straße, 35043 Marburg, Germany; lisastuckenschneider@gmail.com; 2Fachbereich Chemie, Philipps-Universität Marburg, Hans-Meerwein-Straße 4, 35032 Marburg, Germany

**Keywords:** bacterial cell wall synthesis, penicillin-binding proteins, *Bacillus subtilis*, single molecule dynamics

## Abstract

We have used single molecule tracking to investigate dynamics of four penicillin-binding proteins (PBPs) in *Bacillus subtilis* to shed light on their possible modes of action. We show that Pbp2a, Pbp3, Pbp4, and Pbp4a, when expressed at very low levels, show at least two distinct states of mobility: a state of slow motion, likely representing molecules involved in cell wall synthesis, and a mode of fast motion, likely representing freely diffusing molecules. Except for Pbp4, all other PBPs showed about 50% molecules in the slow mobility state, suggesting that roughly half of all molecules are engaged in a substrate-bound mode. We observed similar coefficients for the slow mobility state for Pbp4 and Pbp4a on the one hand, and for Pbp2a and Pbp3 on the other hand, indicating possible joint activities, respectively. Upon induction of osmotic stress, Pbp2a and Pbp4a changed from a pattern of localization mostly at the lateral cell membrane to also include localization at the septum, revealing that sites of preferred positioning for these two PBPs can be modified during stress conditions. While Pbp3 became more dynamic after induction of osmotic stress, Pbp4 became more static, showing that PBPs reacted markedly differently to envelope stress conditions. The data suggest that PBPs could take over functions in cell wall synthesis during different stress conditions, increasing the resilience of cell wall homeostasis in different environmental conditions. All PBPs lost their respective localization pattern after the addition of vancomycin or penicillin G, indicating that patterns largely depend on substrate availability. Our findings show that PBPs rapidly alter between non-targeted motion through the cell membrane and capture at sites of active cell wall synthesis, most likely guided by complex formation with other cell wall synthesis enzymes.

## 1. Introduction

Bacteria display strikingly different shapes (coccoid, rod shaped, or higher degrees of morphologies). Cell morphologies are maintained and determined by the cell wall, a sacculus of so-called murein, or peptidoglycan (PG). The rigid cell wall is built up of single or multiple layers, with a thickness of ~3–6 nm for Gram-negative and 10–40 nm for Gram-positive bacteria [1]. The cell wall not only maintains cell shape, but also renders cells more resilient against environmental influences such as osmotic and ionic changes, which can lead to a drastic turgor change inside the cell [1,2,3,4]. The cell wall layers consist of peptidoglycan chains interconnected by peptide bridges to build up a mesh-like, strong sacculus. PG polymer chains may have variable lengths and are composed of N-acetylglucosamine (GlcNAc) and N-acetylmuramic acid (MurNAc) disaccharide subunits. The pentapeptide side chain, containing L- and D-amino acids, is attached to the MurNAc part of the disaccharide subunit [1,4].

The bacterial cell is a growing and dividing entity, thus, the peptidoglycan layers need to grow concomitantly by the (lateral) insertion of new PG chains into the sacculus [2,4]. Essential players in this process comprise penicillin-binding proteins (PBPs) and glycosyltransferases RodA and FtsW [2,4]. The *Bacillus subtilis* genome contains 16 genes categorized as encoding PBPs, which have redundancies in their enzymatic activities. PBPs can be subdivided into four classes. Class A PBPs are high molecular weight proteins with dual functionality, with transglycosylase (TG) and transpeptidase (TP) activity for the elongation of PG strands and their crosslinking, respectively. Class B PBPs are monofunctional high weight proteins with TP activity, i.e., catalyzing the crosslinking of the peptide side chains. Additionally, there are two low molecular weight proteins classes of PBPs with two different activities, carboxypeptidase and endopeptidase, that are required for maintaining the correct level of PG crosslinking, which is specific for each bacterial species [1,4,5,6,7,8].

Class A PBPs were recently shown to confer a role in cell wall repair, while class B PBPs cooperate with RodA in lateral cell wall extension in *Escherichia coli* [9]. This would explain the redundancy of some PBPs. Also, several PBPs were shown to be involved in cell division rather than in cell elongation, and logically localize to division sites rather than to the lateral sides of rod shaped (or curved) cells [5,10,11,12]. In *B. subtilis*, an equilibrium between the Rod complex (RodA, Pbp2a, or PbpH plus MreBCD) and class A PBPs was shown to be responsible for the maintenance of proper cell diameter [13]. However, many of the functions of multiple PBPs are still unclear. Therefore, we aimed at further understanding the dynamic behavior of *B. subtilis* PBPs during exponential growth, and under conditions of cell wall stress, in order to gain a more detailed insight into the three-dimensional organization of cell wall synthesis and maintenance. We used single molecule tracking (SMT) to determine and quantify molecules with different mobilities of four different PBPs, assuming that slow-moving ones are involved in the insertion of peptidoglycan, during cell growth and under different types of stress (osmotic and antibiotic). We analyzed one class A, two class B PBPs, and one PBP with carboxypeptidase activity. According to in vivo observation, PBPs are thought to travel along paths perpendicular to the long axis of cells, together with enzymes extending the PG strands, as well as with MreB [14,15,16]; some PBPs were shown to directly interact with MreB [12]. Mobilities of MreB and of directionally moving PBPs were determined to be in the range of 20 to 50 nm/s [13,17,18,19], which, when using SMT visualization, is characterized by low mobility, in contrast to free diffusion, which results in much higher diffusion constants of molecules [11,20]. Because freely diffusing molecules cannot be captured by conventional epifluorescence or confocal scanning microscopy, we analyzed PBP mobilities using SMT, which captures images in the range of 10 to 30 ms, rather than several hundred milliseconds or minutes. We found similarities in movement of PBPs, indicating possible cooperation in activities, but also differences, especially during stress conditions. In such conditions, the localization and modes of motion showed considerable changes, indicating that different PBPs may play different roles during adaptation to cell wall stress, possibly by reorganizing cell wall synthesis, or by simply adapting to different conditions within the cell envelope. Importantly, we found major changes in localization patterns rather than molecular dynamics following treatment with antibiotics, revealing that substrate availability plays a major role in finding sites of activity for PBPs, besides binding to MreB filaments.

## 2. Materials and Methods

### 2.1. Strain Preparation

All strains were produced with a modified pHJDS vector [21], in which the *gfp* gene was replaced by the *mVenus* gene by Gibson assembly to generate N-terminal fusion proteins with mVenus. pHJDS is an integration vector to insert the fusion protein into the original locus on the genome by single crossover homologous recombination. pHJDS contains two resistance cassettes (a *bla* and *cat* gene). Plasmids were constructed with the help of Gibson Assembly (Gibson-Assembly^®^ Master Mix Hifi, New England BioLabs, Ipswich, MA, USA) to ligate the insert, consisting of the first 500 base pairs of the corresponding gene with the linearized pHJDS vector. The gene IDs are as follows: *pbpA* (Ppb2a, class B PBP) locus tag BSU_25000; *pbpC* (Pbp3, class B PBP) locus tag BSU_04140; *pbpD* (Pbp4, class A PBP) locus ID BSU_31490; *dacC* (Pbp4a, class C PBP) locus ID BSU_18350. The vectors were then introduced into *E. coli* and 100 µg/mL ampicillin LB agar plates were used to select for transformants. The isolated plasmids were checked by sequencing and subsequently used for the transformation of *Bacillus subtilis* PY79 or the deletion strains. The transformants were selected by carrying the chloramphenicol resistance cassette on LB agar plates with 5 µg/mL chloramphenicol.

### 2.2. Cell Cultivation

All strains (Appendix A) were cultivated on LB agar plates or in liquid LB medium with the corresponding antibiotic at 30 °C under aeration for *B. subtilis* PY79 or 37 °C for *E. coli* cultures. All fusions were induced with 0.01% xylose, such that a barely detectable level of protein was generated. For microscopy, *B. subtilis* strains were grown overnight in LB medium containing 5 µg/mL chloramphenicol and were then inoculated into minimal medium (S750) [22], with low intrinsic fluorescence, at an OD600 of 0.1. The cultures were grown until an OD600 of 0.6–0.8 and were then used for microscopy. For the stress assays, 0.5 M NaCl [23], 1 M sorbitol [24], 4 µg/mL vancomycin [25], or 4 µg/mL penicillin G [25,26,27] were added to the culture that were incubated for an additional 30 min.

### 2.3. Sample Preparation for Fluorescence Microscopy

For single molecule tracking, background fluorescence was reduced by cleaning the round slides (25 mm, Paul Marienfeld GmbH & Co. KG, Lauda-Königshofen, Germany) with 1% Helmanex^®^ III (Hellma GmbH & Co. KG, Müllheim im Markgräflerland, Germany) solution for 30 min in a sonication bath. Next, the slides were washed in distilled water 3–4 times before treating them for another 30 min in the sonication bath. To immobilize cells and to keep them under continued growth, agar cover slips were prepared in 10 mm glass slides, Paul Marienfeld GmbH & Co. KG, by adding 1% extra clean agarose to S750 minimal medium. Then, 5 µL of the cell culture were dropped on the slide and an agar slide was put onto the drop of the cell culture.

### 2.4. Single Molecule Tracking Microscopy and Data Processing

For single molecule tracking, a customized slim field microscope (Nikon Eclipse Ti microscope; 100 × oil-immersion objective, NA = 1.49, Nikon, Tokyo, Japan) equipped with a 514 nm laser diode beam line (100 mW power) and an EMCCD camera (ImageEM X2 EM-CCD, Hamamatsu, Hamamatsu, Japan) was used to generate the data. The beam was extended 20-fold, and its central part was focused on the back focal plane of the objective, generating an almost parallel illumination of a circle with roughly 10 µm diameter, with relatively even energy distribution [28]. The energy values were close to 160 W cm^−2^ of power density. All movies were generated with the following settings: 20 ms, 20 mW laser power, and 3000 frames.

For processing the data, three major steps were performed. Firstly, the movies were cropped to the same length of 2001 frames to remove the initial frames not being at signal molecule level. This state can be deduced from a decay curve using SMTracker 1.5. The cropped movies were summed up to have a projection of the fluorescence signal in the cell. For this process the freely available software Fiji [29] was used. The sums and the brightfield pictures were subsequently used for the cell outlines with the help of Oufti [30]. To generate trajectories with a minimum length of 5 frames from the movies, Utrack [31], a Matlab-based program, was used. Finally, data were analyzed using SMTracker software [32]. The figures were generated in and taken from SMTracker 2 software (https://sourceforge.net/projects/singlemoleculetracker/ last update Version 2 25 June 2021) and were modified with the help of Matlab (latest v. R2024b).

### 2.5. Western Blot

The cells were cultivated as for microscopy with 0.5% xylose to have a higher amount of the corresponding fusion protein. Pellets from 50 mL cultures were resuspended in 1 mL of lysis buffer containing 100 mM NaCl, 50 mM EDTA, 0.1 mg/mL RNase, 0.01 mg/mL DNase, and 1.26 mg/mL Lysozyme and were subsequently mixed with 4× SDS sample buffer. Then, 20 µL were loaded on a 10% SDS gel. The mVenus tagged proteins were detected using an anti-GFP antibody and the membrane was treated with Immobilon Forte Western HRP substrate (Merck) to visualize the proteins.

## 3. Results

### 3.1. SMT-Based Determination of the Spatial Pattern of the PBPs

We generated N-terminal mVenus-YFP fusions to Pbp2a, Pbp3, Pbp4, and Pbp4a, which were integrated at the original gene locus, driven by the xylose promoter, and were thereby expressed as the sole source of the protein. For Pbp2a, Pbp3, and Pbp4, the N-terminus is expected to be in the cytosol (Figure 1A). For Pbp4a, it was speculated that the N-terminus may be peripherally membrane-anchored (Figure 1A). Therefore, the N-terminal fusion may cause unnatural orientation of the N-terminus. We found that a low level of induction using 0.01% xylose driving the xylose promoter resulted in cells that had a regular rod shape, similar to that of wild type cells (Figure 1B). Growth experiments of all created mVenus-PBP strains showed that the growth patterns were similar but not identical, although the spread of standard deviations (SDs) did not reveal statistically significant differences in these curves compared to those of wild type *B. subtilis* strain PY79 lacking any fusion construct (Appendix A). Clear bands for the full-length expression of Pbp2a and Pbp4 could be verified by Western blotting (Appendix A) when pxyl promoters were fully induced using 0.5% xylose (Appendix A). Weak bands for mVenus-Pbp3 and mVenus-PbpH (from another study) could be observed above the background of PY79 cells not expressing any fusion, but a similar band was also observed for mVenus-Pbp4a, which was expected at 81 kDa, so the expression of Pbp3 and Pbp4a could not be unambiguously shown.

Functionality was tested by generating strains expressing the mVenus-PBP-fusion carrying deletions of paralogous genes that show altered cell morphology in the case of double or multiple deletions [37,38,39]. ∆*pbpH* cells expressing mVenus-Pbp2a or ∆*pbpG pbpF ponA* triple mutant cells expressing mVenus-Pbp4 showed normal morphology as well as normal growth rates (Appendix A), suggesting that these two fusions can functionally replace wild type proteins. In order to avoid overexpression artifacts, all fusions were expressed at a very low level (0.01% xylose). Thus, with the caveat that we could not prove functionality for the Pbp3 and Pbp4a fluorescent protein fusions, which could also be altered from lipid-anchored to containing a transmembrane domain (Figure 1A and legend to Figure 1), we continued to investigate the dynamics of the few molecules expressed, which is possible using single molecule imaging.

We expected that all PBPs would move within the cell membrane, which was shown for Pbp2 [9,40,41] in *E. coli*, as well as for GTase RodA [42,43]. Conventional fluorescence microscopy has its limits in visualizing (membrane) proteins in vivo because diffusive motion of proteins blurs out distinct signals between frames, with exposure times being in the range of several hundred milliseconds. Therefore, we turned to single molecule tracking (SMT) microscopy to gain further insight into the behavior and dynamics of different classes of PBPs. We used YFP-based SMT [28,44] to avoid the inhibition of growth due to blue light inhibition and obtained the continued growth of the cells. We tracked molecules (signals) using 20 ms stream acquisition (Movie S1) to visualize even freely diffusive molecules. The obtained movie data were processed to obtain molecule tracks (only five or more continuous steps were taken into account) using utrack, and all further data analyses were performed using SMTracker 2 [45]. Keeping in mind the caveat of a possibly reduced functionality, we observed different localization patterns for the four PBPs when all signals from stream acquisitions were overlaid into a single image (“sum image” of frames from SMT): Pbp4 and Pbp4a showed uniform distribution throughout the cell membrane, while Pbp2a and Pbp3 revealed a more heterogeneous localization within the cell membrane (Figure 1B); for equally scaled images, see Appendix A. Scheffers et al. [5] localized all 16 PBPs of *B. subtilis* using epifluorescence, and the patterns observed for Pbp3 and Pbp4 agree with our sums data (Figure 1B). Pbp2a had a more patchy/less uniform distribution in our data sets compared to those observed before, while Pbp4a was more equally distributed within the membrane compared to the more patchy localization from Scheffers et al. [5]. We would argue that our analysis using high speed acquisition and high localization precision (usually below 50 nm) more accurately describes protein localization than conventional fluorescence microscopy. In any event, our data are in qualitatively good agreement with previous studies and reveal membrane localization for all four fusion proteins, including Pbp4a where the nature of the membrane attachment was possibly altered through the mVenus fusion.

### 3.2. PBPs Move as Two Populations with Distinct Mobilities

After localization was analyzed via summing up of the molecule tracks, we moved on to investigate the diffusive behavior of the four PBPs. All are predicated to interact with MreB, based on bacterial two-hybrid analyses and pull-down experiments [12]. For MreB and Pbp2a, the single particle approach of Garner et al. (2011) has shown that components of the PGEM can be separated into two different populations, one moving slowly and directionally, and the other one fast and non-directionally [15]. Similar results were shown by Dersch et al. [11] for RodA, MreB, and PbpH, a redundant transpeptidase of Pbp2a in *B. subtilis* [37]. Interestingly, for Pbp2 in *E. coli*, SMT analysis has also shown that two distinct mobilities for molecules exist, a freely diffusive population, and a slow mobile fraction, the latter of which was suggested to be involved in enzymatic function, i.e., extending and/or crosslinking of newly incorporated cell wall material [17]. We wished to investigate if this is also true for other PBPs from *B. subtilis*.

For a comparison of single molecule dynamics of the different proteins to be analyzed, we first used jump distance (JD) analysis, which is based on squared displacement (SQD) analysis, to quantify the mobility of each of the mVenus-PBP fusions. The obtained data could not be satisfactorily explained using a single Rayleigh fit, but were well explained using two fits (note that SMTracker uses several statistical tests, and R2 values of >0.98 were obtained using two simultaneous fits), indicating the existence of two populations with significantly different diffusion constants (Figure 2A–D). This can be seen in Figure 2A–D, where the sum of two Rayleigh fits can explain the observed jump distances very well. Note that SMTracker uses Bayesian Information Criterion to avoid over-fitting of the data. We deduce from these findings that all four PBPs alternate between a freely diffusive state and a low mobility state, in which they are engaged in cell wall synthesis, similar to what was shown for other PBPs in *B. subtilis* [11,15] and Pbp2 in *E. coli* [17].

In SMTracker, all observed tracks can be projected into a standardized, medium-sized cell of 3 × 1 µm. Note that in this case, tracking was performed focusing at the mid-cell plane. From these “heat maps”, shown as insets in Figure 2A–D, the most probable localization for all molecules detected can be deduced. As expected, the localization was membrane-associated for all four PBPs. For Pbp3 and Pbp4, a tendency to localize at the septal area and cell poles was visible, in accordance with the described localization from Scheffers et al. [5] and Sassine et al. [46]. Pbp2a is one of two known transpeptidases for elongation [15,37], and had an almost equal distribution within the cell membrane, while Pbp4a showed a clear preference for polar regions. Thus, PBPs showed very distinct preferences for localization within the membrane. To most accurately follow the trajectories of molecules along the cell membrane, and thus to most correctly determine diffusion coefficients for SQD analyses, the focal plane was shifted from the middle of the cells to the upper part of the cells. This way, we could track molecules diffusing along the length of the cells as well as those moving along the short axis (i.e., moving perpendicular to the long axis); when the focus is at the cell middle, circumferentially moving molecules would not be efficiently tracked. Except for Pbp4, all PBPs showed almost equal distribution between static/low mobile and mobile molecules. For Pbp2a, static fraction had a diffusion constant D1 = 0.08 µm^2^/s (47.4%) and D2 = 0.57 µm^2^/s (52,6%) for the mobile fraction (Figure 2F and Appendix A). Note that SD values in Appendix A (and Figure 2F) reflect deviation of the observed data from the data modeled for a two-population fit, based on Brownian motion [44], and are close to “0” when the fit explains the observed data very well. These diffusion coefficients are not identical to the velocities known from the literature [14,15] because of different settings during microscopy and different calculations. Similar behavior was observed for Pbp4a (D1 = 0.06 µm^2^/s (49.3%), D2 = 0.54 µm^2^/s (50.7%) (Figure 2F). Note that a diffusion coefficient of 0.5 to 0.6 µm^2^/s is within the range of a freely diffusive membrane protein with a single transmembrane domain [47,48], while that of 0.06 or 0.08 is about ten-fold lower and thus incompatible with free diffusion. These findings suggest that about half of these PBP molecules are diffusing until a binding site is reached, while the other half is engaged in a slow mobile, likely enzymatically active state. Diffusion coefficients of static fractions of the four PBPs, D1, are statistically significantly different among all PBPs (*p* < 0.05) except for Pbp3 and Pbp4a (*p* = 0.11) (Figure 2E,F). The similarity could indicate a possible functional interaction of Pbp3 and Pbp4a in coordinated cell wall synthesis, when we assume that similar slow mobilities mean co-mobility within the enzymatically active mode. D1 of Pbp2a (0.081) and of Pbp4 (0.093) are much closer to each other than to D1 or Pbp3 and Pbp4a; because large data sets (with low SD) tend to show very low p values even if numbers are quite similar between them, a co-activity of Pbp2a and Pbp4 could be possible. It was also claimed that Pbp4 might be interacting with Pbp3 and Pbp4a because of the localization patterns [5], but our tracking data (based on very different D1 between Pbp4 and the other two PBPs) are not supportive of this suggestion. From our analysis, it is more likely that Pbp2a and Pbp4 are moving together, and when considering the biological functions (Pbp2a is a transpeptidase and Pbp4 is a bifunctional glycosyltransferase and transpeptidase), one might consider that a correlation of their diffusion constants may indicate a similar substrate binding and synthesis activity.

### 3.3. The Mobility of the PGEM (Peptidoglycan Elongation Machinery) Transpeptidase Pbp2a Is Influenced by Ionic Stress and the Antibiotic Vancomycin

Based on the current level of knowledge, it is assumed that all the different PBPs are responsible for parts of the peptidoglycan/cell wall synthesis, be it elongation or repair, but it is still unclear why so many PBPs are involved in this process. We wished to investigate how PBPs change their mobility under stress conditions that might have an effect on the cell wall. By analyzing changes in static fractions after cell wall stress, which we assume correspond to changes in substrate binding and the enzymatically active state (or binding to the PGEM, which is less likely), we intended to search for differences in dynamics that could explain the partially overlapping functions of PBPs. We chose 0.5 M NaCl as moderate osmotic as well as ionic stress, 1 M sorbitol as moderate osmotic stress [24], 4 µg/mL vancomycin (which is similar to the commonly used concentration of 2 µg/mL (10 × MIC) [25]), as cell wall inhibitor binding to the D-alanyl-D-alanine of the peptide of nascent peptidoglycan units (inhibiting both transglycosylase and transpeptidase activity) [25,26,27], and 4 µg/mL penicillin G, as a β-lactam inhibiting PBPs activity via binding to the active site mimicking the terminal D-alanyl-D-alanine dipeptide [26,49,50]. Stress conditions were applied at an OD600 of 0.6–0.7 for 30 min before microscopy.

All four stress factors led to a relocalization of Pbp2a compared to the non-stressed condition. The probability to find this PBP at the septal area was higher during osmotic stress conditions (Figure 3B,C), which may indicate a higher substrate availability at the septum during osmotic stress, since it is known that substrate availability can alter the localization of Pbp2a [10]. Contrarily, the addition of vancomycin and penicillin G led to the loss of the regular localization pattern (Figure 3D,E). Note that diffusion coefficients remained constant because of the GMM (Gaussian mixture model) method used to analyze the data (D1 = 0.08 µm^2^/s; D2 = 0.71 µm^2^/s). GMM keeps D at one value calculated for all conditions (five in this case) to better compare changes in population sizes, which are most relevant for our analyses, and thus allow for an easier comparison of different conditions for one PBP. Therefore, D determined from GMM using five different conditions slightly differs from that estimated by JD for exponentially growing cells (Figure 2).

The mobility fractions of Pbp2a changed with NaCl and vancomycin stress application (30 min of treatment) (Movie S2); however, for sorbitol, the populations did not change significantly (Figure 3F). With NaCl stress, the static protein fraction became smaller compared to the non-stressed condition (Figure 3F, Appendix A), which was also shown for the redundant protein PbpH and the GTase RodA [11]. This effect was even more severe when the cells were stressed by vancomycin. Here, the static fraction shifted by 28% from 50% (non-stressed) to 36% (Figure 3E). This is in agreement with reduced substrate availability for Pbp2a in the cell wall by vancomycin, which would also explain the more random localization seen in Figure 3D. It is known that 100 µg/mL of vancomycin slows down the circumferential movement of PbpH, a transpeptidase redundant with Pbp2a [14]. In contrast to vancomycin, penicillin G had no considerable effect on the dynamics of Pbp2a (Figure 3E,F). This might be explained by the phenomenon known from *E. coli*, where different ß-lactams effect different steps of the cell cycle [51]. Penicillin G or benzylpenicillin seem to just affect the division site-associated PBPs, while Pbp2a is known to be part of the elongasome, and it is also known that the *E. coli* homolog is less affected by penicillin G treatment [51]. However, our data suggest that PBPs relocate to the septum during mild stress conditions (Figure 3B,C), hinting towards a possible secondary function of Pbp2a at the septum beside the important one within the PGEM machinery.

### 3.4. Pbp3 Mobility Changes Markedly Under Osmotic Stress and Antibiotic Treatment

Pbp3 is a class B PBP with transpeptidase activity [5,6,46]. It most likely interacts with the divisome, but it was also described to localize along the cell periphery as distinct foci and bands [5,46]. This is in agreement with SMT data, showing that Pbp3 is also enriched away from the division septum (Figure 2B), and also shows a large concentration at mid cell (Figure 2B and Figure 4A). As observed for Pbp2a, we found a shift in the preferred localization of Pbp3 from the lateral sides to the septal area taking place under the applied osmotic stress conditions (Figure 4A–C), indicating that the substrate for PG synthesis might be sensitive to high osmotic pressure conditions and might be available at different sites. A stronger impact was observed upon the addition of vancomycin and penicillin G. This led to a completely altered pattern, where Pbp3 was entirely delocalized within the cell membrane (Figure 4D,E). This suggests that the availability of the substrate plays a major role in the localization for Pbp3 as it is already known for Pbp2a and for PbpH, as well as for two other TPases of *B. subtilis* [10].

While 0.5 M NaCl had a minor effect on the diffusive behavior of the protein (static fraction in non-stressed cells 55.4% or 59.2% after NaCl stress, Figure 4F), the effects of 1 M sorbitol, 4 µg/mL vancomycin, or 4 µg/mL penicillin G were more pronounced. The number of slow-moving molecules decreased by 15.9% (sorbitol stress) or 24.7% (vancomycin stress) or 21.8% (penicillin G stress) compared to the data from the non-stressed cells (Figure 4F). For vancomycin, we had expected that many Pbp3 molecules would shift from a substrate-bound to a mobile mode due to the loss of binding sites as well as for penicillin G, which is in accordance with *E. coli* data stating that penicillin G is influencing the division site (42). For sorbitol, we were surprised to see a large change, because on the other hand, the ionic stress caused by NaCl did not lead to such a marked difference in mobility (Figure 4F). Although we have no explanation for this phenomenon, it is apparent that the two stress conditions result in a considerably different mode of mobility for Pbp3, which we propose reflects changed conditions for cell wall synthesis during different environmental conditions.

As vancomycin forms a complex with the terminal D-Ala-D-Ala of the peptide side chain of the PG and binds to PG precursors [25], it makes sense that the addition of vancomycin led to a higher proportion of unbound mVenus-Pbp3 proteins (as reflected by the decrease in steps around “0” and increase in steps towards ±0.5 µm, as seen in Figure 4D, compared with Figure 4A; in other words, the distribution of steps becomes broader), searching for an interaction site with the peptide side chains of PG strands, similar to Pbp2a. This indicates that substrate availability is an important determinant for PBP mobility, rather than an interaction with other proteins or a protein complex to extend or crosslink PG strands.

### 3.5. Osmotic Stress Leads to an Altered Localization of Pbp4 but Not Substrate Availability Based on Vancomycin Treatment

Pbp4 is a class A PBP with bifunctional activity (TG and TP) [5,6,38,39]. The heat maps of the unstressed conditions correlate with the published localization of Pbp4 (septal localization and some peripheral orientations, Figure 5A) [5]. When cells were stressed with 0.5 M NaCl for 30 min, no severe effect involving the Pbp4 fusion was noticeable. Despite this, the addition of 1 M sorbitol led to a profound increase in the static fraction, by 50.1% (Figure 5C,F, Appendix A). This is in stark contrast to Pbp3, which became more mobile during sorbitol stress (Figure 4C,F). Our findings suggest that during osmotic stress, Pbp3 can find fewer substrate binding sites, and Pbp4 can find more, while osmotic plus ionic stress (NaCl) leaves binding and diffusion patterns relatively unaltered. Sorbitol stress also led to a change in localization of Pbp4 (Figure 5C), and the addition of vancomycin led to complete delocalization (Figure 5D). Note that the heat map in Figure 5D still contains more than 2500 tracks, as compared with close to 3000 tracks in the non-stressed cells (Appendix A), so Pbp4 molecules are still present and moving/arresting, but no longer at any preferred sites within the membrane.

Penicillin G led to an increase in the static fraction by 17.6% (from 39.7% to 46.7%) (Figure 5F) and was thus almost as affected by penicillin G like Pbp3. Pbp4 changed its localization pattern to a more even distribution around the cell. Thus, penicillin G treatment influenced class A and class B PBPs from the division site in a similar manner.

These findings show that slow mobility, in our interpretation, indicating binding to a substrate, continues during inhibition/reduction in peptide crosslinking as well as elongation, while the preferred positioning of the enzyme is less noticeably altered compared to that of the other PBPs. Lages et al. (2013) showed for Pbp1, another class A PBP, and for Pbp2b, a class B PBP, and also a division associated PBP like the investigated Pbp4, that a change in substrate availability does not lead to an altered localization of these PBPs, but that substrate availability has more influence on PBPs with peripheral (non-septal) localization [10]. Because we applied a different spatiotemporal resolution compared to conventional wide field fluorescence microscopy, we might have seen even slight alterations of the localization pattern, caused by vancomycin treatment, or alternatively, the lateral mode of localization has an influence on the dynamics of Pbp4. Thus, the underlying mechanism for why substrate availability is more important for PBPs with a peripheral localization compared to division-associated paralogs still remains unclear.

### 3.6. Pbp4a Motion Is Mainly Affected by Vancomycin and Undergoes Relocalization Under Stress Conditions

Pbp4a (encoded by *dacC* gene) is a D,D-carboxypeptidase, one of two D,D-carboxypeptidases that are involved in vegetative growth [1,5,7,8]. Because of its function as a carboxypeptidase, it contributes to the regulation of cell wall crosslinking by reducing the number of sites that can be crosslinked [6,8]. In the heat map of exponentially growing cells (Figure 6A), a preference for localization at subpolar regions is apparent, as well to sites at the lateral cell wall, but not at the septal region. When the cells were stressed with NaCl or sorbitol, a relocalization of Pbp4a took place towards the septum (Figure 6B,C), similar to what was observed for Pbp2a and for Pbp3. However, the sizes of the mobile and static fractions did not alter considerably (Figure 6B,C). Interestingly, in response to vancomycin and penicillin G treatment, the localization pattern became more diffuse (Figure 6D–F), and 22.1% (vancomycin) or 35.3% (penicillin G) of the molecules lost the static/slow mobile mobility (Figure 6D–F, Appendix A). These data suggest that vancomycin masks Pbp4a binding sites, or that Pbp4a activity is coupled to that of other PBPs, especially of Pbp2a and Pbp3 who also became more mobile during vancomycin treatment (Figure 3D and Figure 4D). Compared to the other PBPs, Pbp4a had an overall larger static fraction, suggesting that two thirds of all Pbp4a molecules are engaged in substrate binding, and its diffusion coefficients (D1 = 0.09 µm^2^/s) in the low mobility mode differed somewhat from that of the other three proteins, while being most similar to that of Pbp2a. These findings suggest that Pbp4a may operate in a mixture of independent movement and PBP-dependent substrate association. Please note that values between SQD and GMM analyses differ because they are independently calculated in SQD but are based on a comparison between different conditions used in GMM; however, the obtained values were in a comparable range.

### 3.7. PBPs Have Different Average Residence Times

By analyzing the average residence time of all molecules, we wanted to understand how long molecules stay in a set radius of 120 nm (about three times our localization error) for nine intervals or longer, and whether this behavior changes with different stress conditions. Of note, our determined values are an underestimation of true dwell times in vivo due to molecule bleaching during acquisitions (average half-life of YFP is about 1200 ms under comparable experimental conditions [32]). Keeping this in mind, we can still use dwell times to compare behavior between proteins that carry the same chromophore and between different conditions. Additionally, while being related, changes in dwell times do not directly reflect alterations seen in static and dynamic populations because slowly moving molecules captured for five to nine steps were not included in the dwell time analyses.

For Pbp2a and Pbp3, no noticeable changes between exponential growth and osmotic stress conditions were visible (Figure 7), but the residence times were increased with vancomycin treatment, and likewise significantly longer with penicillin G treatment. Note that the statistically significant difference between exponential growth and vancomycin stress for Pbp3 is likely due to narrow SD values and high numbers of determined events but does not hold true in terms of its actual significance. The residence time for Pbp3 was marginally lowered under osmotic stress. As population size for the slow mobile mode changed for Pbp3 during sorbitol stress (Figure 4C), but (long) dwell times did not alter markedly, it is evident that once a molecule is in a slow mobile (static) mode, it remains there for roughly the same time, no matter if the cells grow exponentially or if they are stressed. This is different for Pbp4a: this PBP showed noticeable changes in its residence time in response to all stress conditions. With 0.5 M NaCl and 1 M sorbitol, average residence time became somewhat shorter from 0.30 s ± 0.01 s to 0.28 s ± 0.004 s (0.5 M NaCl) and 0.28 s ± 0.004 s (sorbitol). Vancomycin or penicillin G led to a decrease in the residence time from 0.3 s ± 0.011 s to 0.27 s ± 0.0033 s (vancomycin) or 0.27 ± 0.0029 s (penicillin G). Note that these changes were statistically significant (Figure 7). This supports the idea that Pbp4a binding is directly affected by vancomycin, rather than indirectly via the inhibition of Pbp2 or Pbp3, because their dwell times did not change much in response to the inhibition of peptide crosslinking. For Pbp4, residence times decreased during most of the stress conditions, except for the sorbitol stress where the residence time increased (Figure 7). For all four PBPs, residence time changed the most after penicillin G treatment, especially for Pbp2a, which is remarkable because its mobility did not change after penicillin G treatment. 

Interestingly, Pbp4 and Pbp4a showed significantly longer dwell times than Pbp2a and Pbp3, indicating that enzymatic activity of Pbp2a and Pbp3 may not be directly coordinated with that of the other two PBPs, which we proposed above based on similar diffusion coefficients for Pbp2a and Pbp4, and Pbp3 and Pbp4a, respectively. Coordination would be based on joint circumferential movement of PBPs together with TG enzymes [13,42]. Thus, Pbp2a, Pbp3, and Pbp4 showed relatively robust residence times, and therefore apparently robust enzymatic activities during stress conditions, while those of Pbp4a were affected by stress. Different dwell times of PBPs support the idea of dynamic associations/dissociations rather than coordinated formation and the dissociation of stable protein complexes.

## 4. Discussion

Peptidoglycan synthesis is a hallmark of bacterial life style, and therefore a major drug target. Synthesis is performed by a large array of proteins, from sugar polymerases to endopeptidases, from transpeptidases to hydrolases. *Bacillus subtilis* PBPs can be mono- and bifunctional, and likely have redundant enzymatic capabilities, or may share labor when it comes to adaptation to different environmental conditions. We have investigated the dynamic behavior of four *B. subtilis* PBPs with different domains/activities, at a single molecule level; all major changes are stated in Table 1.

An important finding is our observation that that inhibition of binding to PG precursors, or of peptide crosslinking, by the addition of vancomycin or penicillin G, respectively, leads to a loss of preferred localization patterns for all four PBPs (Table 1). This finding suggests that substrate availability is an important determinant for preferred PBP locations (keeping in mind that several PBPs are recruited to the septal region through direct protein interactions [51]). During masking of D-Ala-D-Ala sites and loss of precursors, PBPs become more mobile in the cases of Pbp2a and Pbp3, as well as for CPase Pbp4a. For penicillin G, Pbp2a was the only PBP that did not show any changes between slow and high mobility fractions of molecules. Our experiments also suggest that the toxicity of penicillin G is not caused by the strong inhibition of individual PBPs but is likely based on the moderate but additive inhibition of many PBPs. It could be argued that the inhibition of cell wall synthesis affects PBP localization and dynamics because this treatment leads to a reduction in MreB rotation [14,15,16]. However, 100 µg/mL of vancomycin was shown to slow down the circumferential movement of MreB or PbpH (a transpeptidase redundant to Pbp2a [14]), while only 4 µg/mL was used in our studies, suggesting that the loss of preferred localization is not due to an indirect effect via MreB motion.

A second major result from this work is the strong changes monitored for single molecule motion following osmotic stress (Table 1). While changes in the motion of cytosolic proteins following the upshift of ionic conditions were studied to some degree [52], it is not well understood how membrane proteins respond to such changes. Osmotic stress is one of the most common challenges in the natural environment, so soil bacteria need to be especially well prepared to deal with osmotic fluctuations [23]. For turgor adaptation, the cell wall plays a key role, besides membrane-integral ion and amino acid transporters [53]. Interestingly, Pbp2a reacted strongly towards sodium chloride stress, which triggers osmotic as well as ionic stress in the cell, whereas Pbp3 and Pbp4 showed considerable but opposing changes when sorbitol was added, generating non-ionic osmotic stress. Conversely, Pbp4a underwent relocalization under osmotic stress, but no significant changes were observed in the diffusive coefficient of the slow mobile population. Thus, no common but individual alterations in single molecule motion were observed. The caveat in these experiments is a possible artifact of fluorescent protein-tagging of Pbp4, which has to be kept in mind. Interestingly, MurG, an enzyme generating cell wall precursors in the cytosol, is polarly localized in Mycobacteria, together with TG RodA and other PBPs. Actinobacteria grow by polar cell wall synthesis, as opposed to lateral cell wall synthesis such as in *E. coli* or *B. subtilis* [54,55]. When cell wall damage is induced by the addition of cell wall-degrading enzymes into the medium, cell wall synthesis relocates to the lateral sides, and MurG also relocates to sites all around the cell periphery, showing that even intracellular enzymes associated with cell wall synthesis show the adaptation of localization in response to external perturbations [56].

The most plausible explanation for the two distinct modes of diffusion we have seen for all four PBPs is free diffusion in the membrane and low mobility due to engagement in active synthesis. The changes in single molecule motion we observed following osmotic changes might indicate different engagement in cell wall synthesis during strass adaptation. This would agree with experiments from Peters et al. (2016), who showed that Pbp6b is a specialized D,D-CPase in *E. coli* that contributes to cell shape maintenance at low pH, suggesting that the redundancy of D,D-CPases may play an important role in the stress response [57]. Likewise, PBP1a/1b and PBP4 are both bifunctional (class A), and during alkaline stress, PBP1b becomes active while PBP1a and PBP4 lose activity, showing different functions/activities during pH stress [34]. Additionally, it is known that Pbp4* (encoded by gene *pbpE*), a D,D-Endopeptidase, is involved in the high salt adaption of *B. subtilis* [27]. Interestingly, following osmotic stress, Pbp2a and Pbp3 showed relocalization to the septum, suggesting that under changing conditions, the sites of activity for PBPs can be different, and that the mode of synthesis of peptidoglycan is altered under stress conditions. Indeed, Dion et al. (2019) showed that a balance between the two systems, the Rod and the A-PBP system, is needed to maintain the rod shape in *B. subtilis* and that under hyperosmotic conditions, the Rod system (including Pbp2a) is important for reinforcing the cell wall [13].

This work analyzing the diffusion coefficients, population sizes, and preferred localization patterns shows a clear correlation between Pbp3 and Pbp4a, indicating that PG transpeptidases and carboxypeptidases maybe have coordinated activities. We further found that Pbp2a and Pbp4 show a quantitatively similar dynamic behavior. This correlation could indicate a similar coordinated binding to their substrates. Because Pbp4 has a structure homologous to that of Pbp1b of *E. coli*, which was recently shown to have a PG repair function [9], it would be interesting to test if Pbp4 of *B. subtilis* has a similar function.

Our data showing two distinct mobility fractions for PBPs are in complete agreement with findings on PbpH and RodA in *B. subtilis*, and with Pbp2 in *E. coli*, which change between a freely diffusive and a substrate-bound manner, thus dynamically exchanging with the PGEM complex [11,41], rather than being predominantly bound to a slow-moving PG synthesis complex. For *E. coli* Pbp2, it was recently shown that an entirely static as well as a slow mobile fraction exist, besides a freely diffusive one, suggesting two different “active” states, one that is based on substrate binding and one arising from slow extension of the PG strands [17]. We have not investigated if PBPs might have two distinct slow mobile speeds in *B. subtilis* as our relatively fast acquisition speed does not allow for this distinction. Our goal was to use a simpler two state model in order to better compare the dynamics of four different PBPs. However, it will be interesting to determine if distinct “substrate binding” (entirely static) and “actively synthesizing” (slow mobile) fractions exist in Gram-positive cell walls in future work. In earlier studies, PBP dynamics (looking at PbpH) were shown to be strongly reduced after vancomycin treatment (100 µg/mL for 8 min), with regard to ensemble movement of many molecules in TIRF mode, using acquisition intervals in the seconds range [14]. Using SMT and 20 ms intervals, we can show that this effect is based in part on fewer PBPs being in a slow-movement mode, with a diffusion constant of close to 0.1 µm^2^/s, roughly corresponding to the speed of directed movement found for PbpH, RodA, and MreB [11,14,15].

The inhibition of cell wall synthesis is still one of the most prominent targets for constraining bacterial cell growth and infections. The mechanisms for how antibiotics work in the cell are well understood, but the influence on the inhibited proteins in terms of localization and diffusion are less clear. For the purpose of this study, we analyzed the influence of vancomycin and penicillin on four PBPs, which all showed a disturbed localization pattern and increased mobile fraction (except for Pbp4). It will be interesting to study the effect of inhibiting other aspects of cell wall synthesis on the dynamics of synthetic enzymes and further dissect special requirements for enzymes during stress conditions.

## 5. Conclusions

The functions of proteins involved in cell wall synthesis were investigated at a genetic and biochemical level; their dynamics in time and space are still poorly understood. We have followed the dynamics of four penicillin-binding proteins using single molecule tracking to find that their mobility can be best described assuming two distinct populations, a freely diffusive or a static, substrate-bound state. The preferred sites of localization and population sizes changed during stress adaptation, and even more during the addition of antibiotics disturbing substrate binding or enzyme activity. Our data support the idea that cell wall synthesis is driven by diffusion/capture interactions of synthesis enzymes, that sites of activity are dependent on substrate availability, and that PBP dynamics adapt under stress conditions, reflecting specialized function under different environmental conditions. Two-color SMT experiments could help to reveal which PBPs transiently interact with each other within different or the same synthesis complexes. Our findings support the tug of war model for MreB function [18,19]: cell wall synthesis is driven by the coordinated activity of the Rod complex and possibly of various bifunctional PBPs, who bind to and extend sites of substrate availability, dragging along MreB filaments in both orientations perpendicular to the cell’s short axis. If several extending machineries running in the opposite direction bind to the same MreB filament, enzyme motion and thus activity are blocked. This ensures that cell wall extension is not performed by too many enzymes at a given site, but that enzymes are stochastically distributed all over the cell membrane.

## Figures and Tables

**Figure 1 cells-14-00429-f001:**
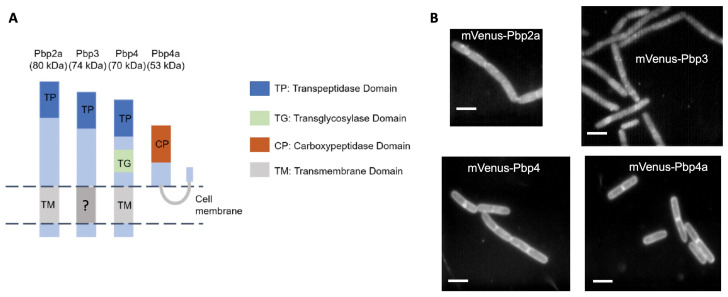
(**A**) shows a schematic cartoon of the targeted four PBPs of *Bacillus subtilis* and their enzymatic domains [33,34,35]. Note that Pbp3 has only a weak prediction for a transmembrane α helix (TM) [36], and a predicted lipoprotein signal peptidase site following the first 20 amino acids, so it could be lipid anchored; this uncertainty is indicated by the question mark. Pbp4a does not have a predicted TM, but a signal peptidase sequence; therefore, a lipid anchor is indicated in the cartoon. Thus, for both Pbp3 and Pbp4a, the N-terminal fluorescent protein (mVenus) fused to the PBP could alter the mode of membrane attachment. (**B**) shows the sum images of the single molecule tracking data of FP-PBP fusions. Scale bars: 3 µm.

**Figure 2 cells-14-00429-f002:**
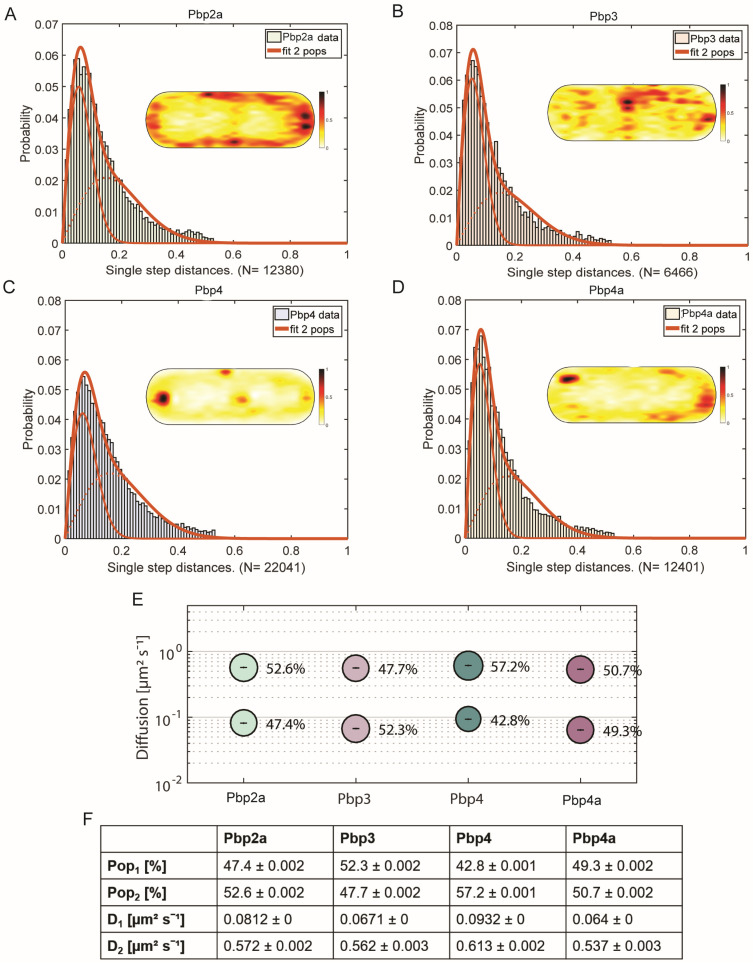
Analysis of PBP dynamics and localization. (**A**–**D**) show jump distance fit of SMT data of four different PBPs in comparison and heat maps. (**E**) Bubble plots of two diffusive populations of the four different PBPs. (**F**) Data table.

**Figure 3 cells-14-00429-f003:**
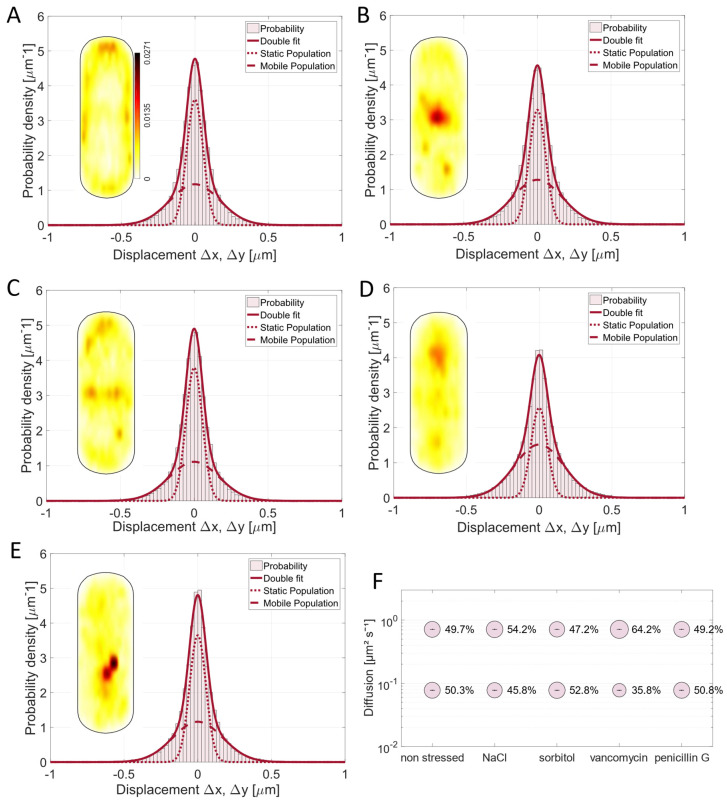
Analyses of mV-Pbp2a dynamics. (**A**–**E**) Gaussian mixture model (GMM) fit of mV-Pbp2a is shown for (**A**) non-stressed, (**B**) 0.5 M NaCl, (**C**) 1 M sorbitol, (**D**) 4 µg/mL vancomycin, and (**E**) 4 µg/mL penicillin G conditions (30 min of treatment) indicating that two-population fit sufficiently explains measured data. For each condition, probability heat map of localization is included as inset, showing different localization patterns of mV-Pbp2a. (**A**) contains scale for all heat maps. (**F**) Bubble plot showing size of population in % and diffusion coefficients for Pbp2a mobility fractions.

**Figure 4 cells-14-00429-f004:**
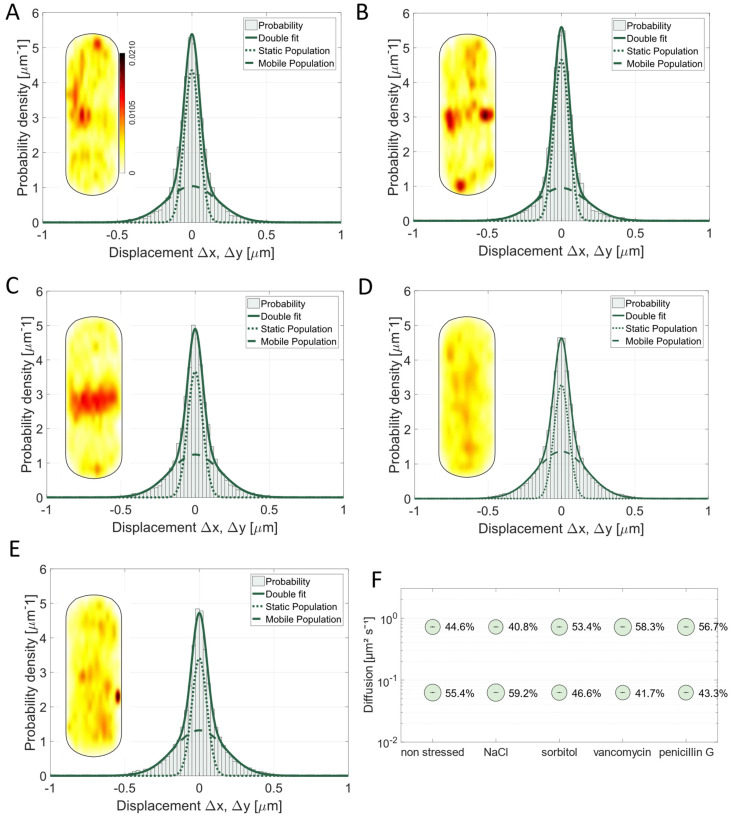
Gaussian mixture model fit of Pbp3 is shown for cells growing (**A**) non-stressed, (**B**) 30 min after addition of 0.5 M NaCl, (**C**) of 1 M sorbitol, (**D**) of 4 µg/mL vancomycin, or (**E**) of 4 µg/mL penicillin G. Two-population fit was sufficient to explain data sets. For each condition, probability heat map of localization is included, showing different localization patterns of mV-Pbp3. (**A**) contains scale for all heat maps. (**F**) Bubble plot showing size of populations in % and corresponding diffusion coefficients for Pbp3.

**Figure 5 cells-14-00429-f005:**
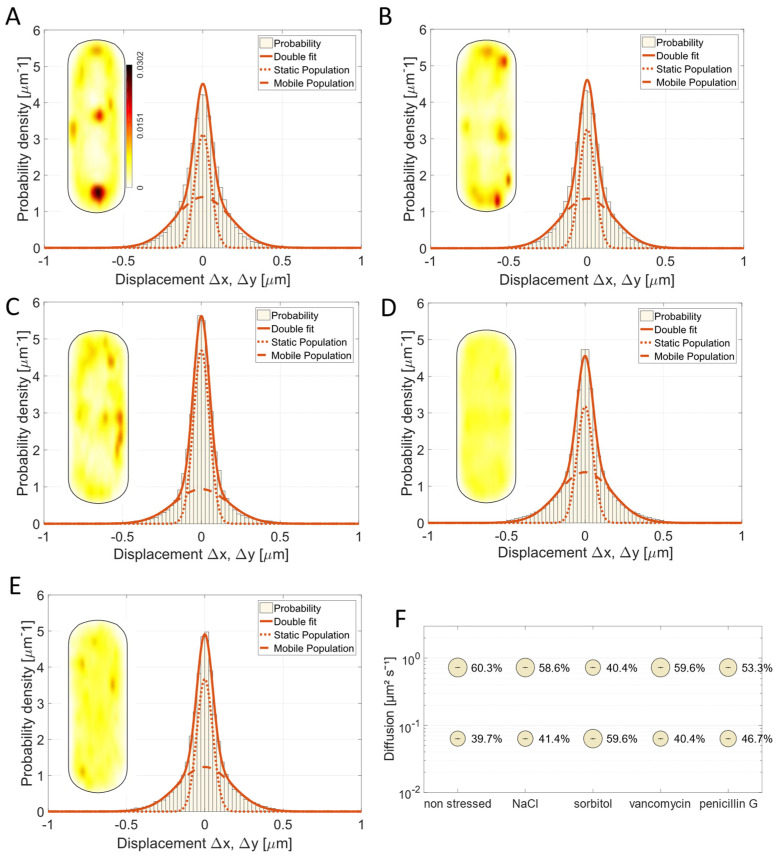
Gaussian mixture model fit of Pbp4 is shown for (**A**) non-stressed, (**B**) 0.5 M NaCl, (**C**) 1 M sorbitol, (**D**) 4 µg/mL vancomycin, or (**E**) 4 µg/mL penicillin G conditions, using two-population fit. For each condition, probability heat map of localization is included as inset, showing different localization patterns of mV-Pbp4. Panel A contains scale for all heat maps. (**F**) Bubble plot showing size of populations in % and corresponding diffusion coefficients of Pbp4.

**Figure 6 cells-14-00429-f006:**
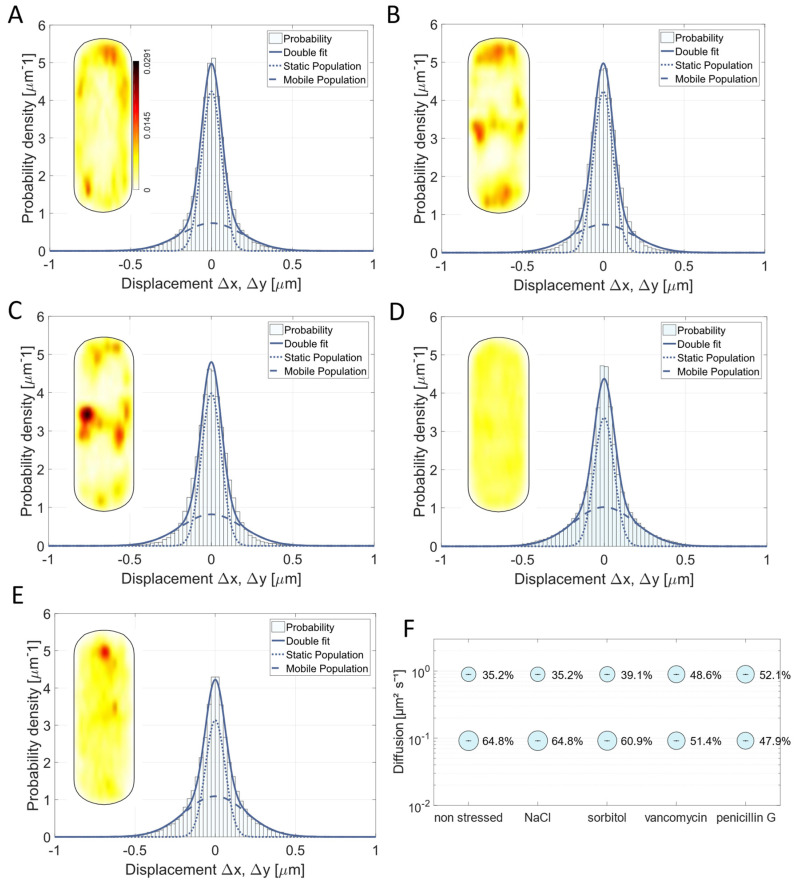
Single molecule dynamics of mVenus-Pbp4a. (**A**–**E**) Gaussian mixture model fit of Pbp4a, shown for non-stressed, 0.5 M NaCl, 1 M sorbitol, 4 µg/mL vancomycin, and 4 µg/mL penicillin G conditions indicating that two-population fit is sufficient for data set. For each condition, probability heat map of localization is included, showing different localization patterns of mV-Pbp4a. (**A**) contains scale for all heat maps. (**F**) Bubble plot showing size of population in % and diffusion coefficients of Pbp4a.

**Figure 7 cells-14-00429-f007:**
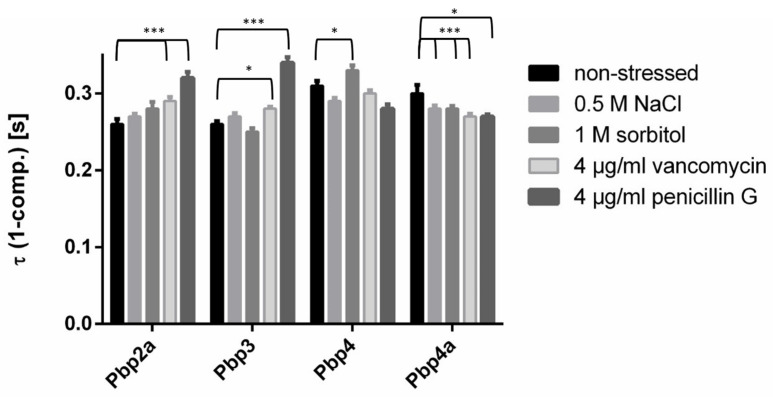
Dwell times of all four PBPs under applied conditions, as judged by events where molecules stay for nine or more consecutive steps within a limited radius. Pbp4 and Pbp4a are staying within a radius of 120 nm slightly longer than Pbp2a and Pbp3. * and *** indicate a *p*-value lower than 0.1 and 0.01 by the significance level of conditions to each other, calculated by a Levene test (Variances of the τ (1-comp.)).

**Table 1 cells-14-00429-t001:** Changes in localization and mobilities in different conditions.

	Localization	Salt	Sorbitol	Vancomycin	Penicillin
Class A					
Pbp4	polar	-	peripheral	dispersed	dispersed
	40% static	- *	more static *	-	more static
Pbp3	peripheral	more septal	more septal	dispersed	dispersed
	55% static	more static	more diffusive	more diffusive	more diffusive
Class B					
Pbp2a	peripheral	more septal	more septal	delocalized	delocalized
	50% static	-	-	more diffusive	-
Class C					
Pbp4a	polar	-	septal	dispersed	dispersed
	65% static	-	-	more diffusive	more diffusive

* Changes in population size was stated above 20% relative difference.

## Data Availability

All raw data are available upon reasonable request from the authors.

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
