# Peer review of "Localization and Single Molecule Dynamics of Bacillus subtilis Penicillin-Binding Proteins Depend on Substrate Availability and Are Affected by Stress Conditions"

_cells, 2025, doi:10.3390/cells14060429_

Round 1
Reviewer 1 Report
Comments and Suggestions for Authors
Review Report
The authors are presenting the study of dynamic localization of 4 different Penicillin Binding Proteins of Bacillus subtilis among the 16 existing in this bacterium. Their aim is to better understand their possible modes of action and 3D organization during cell wall synthesis and maintenance, in normal growth conditions or under cell wall stresses (osmotic and/or ionic pressure, addition of antibiotics targeting the peptidoglycan synthesis). They chose PBPs from different classes (A, B and carboxypeptidase) and a cutting-edge microscopy method; single molecule tracking of fluorescent fusion proteins.
In this way, they show that despite a common membrane localization, each has specific localization characteristics that are affected differently by stresses on the cell wall. In addition, the molecules of each protein separate into two populations with distinct mobility dynamics (slow motion and freely diffusible), which are also modified differently for each PBP depending on the stress conditions.
The approaches used to address this biological question seem appropriate to the objectives and show that different PBPs can play different roles during adaptation to cell wall stress, possibly by reorganizing cell wall synthesis, or simply by adapting to different protein organization within the cell envelope.
General comments :
The manuscript is clear and presented in a well-structured manner with appropriate references.
The experimental design appears appropriate to test the hypothesis and the figures are generally clear, although more detailed comments will be added later.
However, although movies are mentioned in the text, they have not been added to the documents submitted to the journal and therefore cannot be analyzed by the reviewers.
The overall scientific context and objectives are well presented, but I would have appreciated that the authors had given a more detailed description of what is known about each of the PBPs chosen and why they chose them from among the 16 possible.
Please italicize all bacterial names and genes throughout the text, tables and figure legends.
Specific comments :
- Line 32 : the word ‘wall’ is missing, and this is necessary to understand the sentence.
- Please add references for data presented line 61.
- Lines 158 to 163 : To test whether the expression level of fusion proteins with 0.01% xylose is different from that of native proteins, quantitative PCR experiments could be carried out. However, as this does not give direct access to the quantity of protein produced, this could have been supplemented by bocillin staining tests of PBPs in WT strain compared to strains producing the YFP-fusions and possibly to the corresponding pbp-deleted strains to better identify the band corresponding to the protein of interest. One or both of these experiments could be added as additional data.
- Fig.1 : The images shown in B must all be at the same scale to facilitate observation and comparison. Please correct them.
- Fig. S2 : Sample 5 for Pbp4a appears to have been indicated between 2 tracks of the gel. Was it loaded twice? If not, what do these 2 tracks contain? Please clarify the description of this Western Blot.
- Fig. S3 : Same comment as for Fig.1B. Please ensure that all images are to the same scale so that cell sizes can be compared.
- Line 177 : Movie 1 not included in submission. Thanks to add it.
- Fig. 2F and Table S2 : The standard deviation indicated for D1 is equal to 0, which seems unlikely to me. Please correct or explain.
- Lines 253-254 : The differences in D1 coefficients between the PBPs seem very small to me and the standard deviations presented are 0. Please calculate whether the values are significantly different. Similarly for D2, please calculate whether the differences are statistically significant or not.
- Line 299 : Movie 2 not included in submission. Thanks to add it.
- Fig.3 : Thanks to specifically specified the conditions for A, B, C, D and E in the legend.
- Fig. 4 and 5 : Figures 4 and 5 appear to have been reversed. The values in figure 4F are those in table S5 for pbp4 and vice versa. The values in figure 5F are those in table S4 for Pbp3. Please correct this as it greatly complicates the reading and interpretation of the results.
- Line 361 : Please explain what you mean by “ by the increase in longer steps seen in Fig. 4D, compared with 4A”.
- Discussion : The analysis of the diffusion coefficients (D1 and D2) of the Pbp molecules shows a correlation between Pbp4 and Pbp2a and between Pbp3 and Pbp4a. However, the calculation of average residence times shows a correlation between Pbp4 and Pbp4A. If these coefficients give an indication of the active state of the proteins in PG synthesis, how do you explain these differences?
: It would have been interesting to interpret the results according to the class of PBP studied and its known role in PG synthesis. What are the similarities and differences in dynamics and responses to envelope stress between each of them? An additional paragraph for discussion would be a plus.
Reviewer 2 Report
Comments and Suggestions for Authors
The authors employed single molecule tracking (SMT) to map the movement of four penicillin-binding proteins (PBP) in B. subtilis cells in mid-exponential cultures under normal growth conditions or under conditions of stress (osmotic stress caused by NaCl or sorbitol, and exposure to antibiotics which impact on peptidoglycan synthesis, penicillin G and vancomycin). The proteins were expressed under minimal induction of a xylose promotor, to avoid potential impacts of high expression of membrane proteins, and were tracked through epifluorescence (was this GFP or YFP?); both properties were introduced in transformants by homologous recombination at the N-terminal of the proteins. The authors verified expression of three of the four fusion proteins by Western blotting but PBP4a was not confirmed. Evidence of function of two of the four PBP in genetic backgrounds where similar functions were deleted was provided (PBP2a and 4). Despite possible low level of expression, SMT allowed tracking of the few molecules that were expressed so the data for all four PBPs is reported for normal or stress growth conditions. The authors correctly acknowledge the limitations of this approach in several places in the text (‘caveat’ is used three times), acknowledging that expression of the resulting fusion proteins may cause changes in activity, interaction with partners involved in septation or cell elongation (including restricting conformational changes essential for enzymatic function and/or failure to partner with other components of complexes) or prevent correct location in the cell membrane – as documented in the literature and why alternative approaches to studying the location of cell wall enzymes have been developed. Similar tagging approaches have been used for more than 20 years, often not with membrane proteins and recording data with alternative microscopy, and have led to significant discoveries concerning the role of proteins in septation, cell division and peptidoglycan turnover during growth and elongation (referred to as PGEM). The authors argue that SMT is a superior approach and the senior author has several publications using this methodology to explore other components of PGEM and septation, plus localisation of other cellular components/enzymes at the cell membrane. The authors also argue in the Introduction that the roles of many of the proteins involved in cell division and PGEM are not fully defined, although others have published data on the same PBPs explored in this submission, in B. subtilis and other species (see review https://doi.org/10.1016/j.mib.2024.102490 and others in mycobacteria, which lack some of the scaffold proteins detected in other genera).
Overall, the manuscript provides new information on PBPs location, diffusion constants and possibly dwell/residence times (noting the caveat of data limitations for the last, line 433) for normal and stressed conditions, under the experimental conditions used, and allows comparisons with prior publications that employed different microscopical or methodological approaches.
Notwithstanding the above comments, there are several concerns about the manuscript which require addressing:
· some of these are phraseological or editorial (requiring clarification or simply editing to bring the text up to an acceptable standard – italics for Latin words, including bacterial names; using abbreviations for genera once spelt out once; inconsistent styles in References; abbreviations not spelt out at all)
· others are more significant (e.g. what was the vector used? The reference given is for creating insertional mutants tagged with Green Fluorescent Protein, but the text states that mVenus was used as the tag. A reference is required so the nature of the vector used is clear or, if this is new and first use in this submission, then a fuller description is required. Anti-GFP antibodies were used later, Fig. S2, to detect full reads of the insertional mutants but the sizes of the proteins in the Western blot image cannot be interpreted unless the nature of the vector is known so the size of the proteins made can be interpreted as valid or not. Epifluorescence excitation is also noted to be undertaken in a way that growth would not be affected – so was this YFP excitation? Yes, but no obvious until the Results section).
· some issues are technical, some of which will be addressed below, and there is a tendency to accept models from E. coli or other species which are simply hypotheses. The latter is important, as one of the conclusions drawn from this study is that availability of substrate influences the distribution of some of the PBPs studied, which is not proven in the submission and may be incorrect given the action of the antibiotics used – vancomycin binds to substrate, so would limit substrate availability, whereas penicillin binds to enzymes, inhibiting activity (even if substrate is available?).
An annotated pdf is provided to point out some of the concerns raised above.
Abstract
Revise in context of the comments made, particularly drawing conclusions about substrate availability impacting PBP location/movement. There is an underlying assumption that immobile PBP are engaged in PG synthesis or turnover, which is reasonable, but this could be due to substrate binding or stalled machinery, as put forward by authors prior to 2013 (see ref. 41).
Introduction
· all abbreviations require spelling out in full (e.g. TIRF) and make sure that all statements are substantiated with a reference/s, particularly when arguing the superiority of SMT and limitations of other similar methods (e.g. lines 70-72).
· Fig. 1A is mentioned in the Introduction but this contains unreferenced/sourced information and 1B shows a result. Are the authors simply trying to indicate that there is considerable structural information available about the proteins and their location at or in the cell membrane? Please consider what is important to include as background information.
It is noted that the gene locus numbers are not provided in the Methods or in Fig. 1 so it is not possible to seek further information through available databases. Please include this information in the text somewhere, as occurred in a recent Cells article by the senior author for other proteins/genes – which was very helpful.
· Lines 71-80: It would be helpful to have some further background on how SMT has been applied to investigating stress responses in Gram positive cells, especially if this is not commonly used experimentally. This section of the Introduction can be improved in terms of how SMT has been used, on which tagged proteins and what was the experimental approach to be used in this study. It appears that the approaches used were: using SMT to locate the tagged proteins during normal growth; measuring their mobility in normal and stress conditions, noting the proportion of mobile and static populations (is this new?); determining diffusion coefficients (is this new?); and determining dwell or residence times (is this new?). Currently, this part of the Introduction is reporting conclusions which may, or may not, be substantiated from the data presented.
Some of the information in lines 170-180 could be included in the Introduction. Please review the Results section to determine which information would be valuable in the Introduction (e.g. SMT detecting low-level expression against other microscopic methods or protein tagging approaches; risks in using N-terminal tagging for membrane proteins which form complexes with others or where function is regulated by conformational changes which may be precluded by tagging with GFP or YFP).
Also, prior observations and hypotheses around substrate binding determining PBPs locations is not mentioned in the Introduction – the main reference is [41], which contains data published in 2013 on PBP2a, which is studied in the current submission. Ref. 41 should be mentioned, possibly in terms of the limitation of the experimental design (collapse of PMF using nisin, which may have influenced membrane protein mobility; not detecting freely diffusing proteins, if this was the case?). Ref. 41 is of particular relevance, as penicillin and vancomycin were used to test redistribution of PBP, as used in the current submission.
Materials and Methods
· 2.1: Table S1 contains the strain list and should be mentioned here. The vector used for strain construction is cited as ‘this study’, so a map of this vector should be provided even if based on the early reference [18].
· 2.2: line 100 mentions the minimal medium. A reference is needed, particularly when prior publications mention this medium was supplemented with amino acids. What was the C source? Details needed or reference. How was induction performed?
· Lines 102-103: later in the text, the stress conditions used have references, which should occur here. It is important that prior research has shown that the conditions used did, indeed, cause stress.
· The M&Ms section does not contain information on how some of the measurements were undertaken. The details are included in the Results or are missing. Please review the placement of some of these details and it was a little surprising to learn about dwell times (and the limitation of the methods used due to half-life of YFP) much later in the text.
Results
3.1
· Line 151, Fig. S1: the text says that the growth rates were not changed but the diagram may not indicate this for Pbp4. Check kinetics and correct as needed if justified (the growth patterns are similar but not identical, although the spread of SD may not verify any visual differences in these curves).
· Fig. S2: the Western blot shows a grossly overloaded set of controls (lanes 6 and 7), and bands of a similar size in lanes 1, 2, 4 and 5. This is meant to confirm that full-length expression of the proteins occurred but the data are not convincing: PBP4a native protein in 53 kDa (Fig. 1) but there is a faint band of around 100 kDa in lane 5 and all lanes contain a prominent band around 50 kDa: where would the fusion protein for PBP4a occur on this gel? What is the prominent 50 kDa band? What does ‘full induction’ (line 152, main text) mean? The authors acknowledge that full-length expression was not proven for Pbp4a: is this spelt out sufficiently?
· Line 157: Fig. S3 shows expression of the PBP under study in a background where homologs are deleted, which is valid, but there is no growth curve data presented, which is implied in the main text. The legend to S3 is also not very clear regarding ‘no additional substrates added’: if the strains were cultured in LB, would the xylose promoter allow expression of the fusion proteins? Please review the legend for S3 for clarity.
· The authors have correctly concluded that the functional PBP3 and 4a fusion proteins were not proven in this study and should be congratulated on providing the evidence to support this: Fig. 1B shows that the low expression was still detected by SMT, assuming that the ‘FP’ was the YFP as annotated in the diagram but not in the legend.
· Line 183: states that images shown in Fig. 1B showed a ‘regular distribution’ for Pbp4 and 4a. Were these proteins largely at the septum and poles, while the other two PBP were dispersed throughout the membrane (as shown in this figure and in later figures, although the ‘heat map’ show low densities for some of the PBP under study due to different method used)?
· Line 185 onward: compares data in this submission with prior data [5]. Was the same strain used and growth conditions/media by Scheffers et al.? would strain variation be expected, in addition to other experimental variations?
· Line 193: yes, membrane attachment may have been altered for Pbp4a but wouldn’t this also apply to all of the fusion proteins, given that the N-terminal regions are cytoplasmic or transmembrane?
3.2
· Lines 237-250: the authors claim that the freely-moving/diffusing fraction of the tagged proteins are ‘searching for substrate’, but the observations made provide no evidence for this. The less mobile fraction are likely engaged in cell wall maintenance/turnover/PG synthesis and are likely interacting with other proteins in complexes, as shown in the literature for protein interactions. What are the other enzymes doing? ‘hunting’ is one option, not being bound in complexes is another, as is not being functionally regulated through binding? Following one model is a little limiting. Recent publications in mycobacteria may support the ‘hunting’ model, in that data shows movement of MurG during cell wall damage (which implies that precursors of PG synthesis are important during repair). The authors may wish to consider this new information, although mycobacteria grow with a different dynamic to B. subtilis.
3.3 – deals with each of the PBP location and proportion of static versus diffusing proteins.
· Line 290: Ref. 41 provides reasonable data that nisin treatment caused changes in 2a's location and argued that coincidence with patches proved substrate binding was involved. Thesw authors also reported that there was some collapse in the PMF, which may influence the level of support for this model against stalled PGEM. Was this publication/model validated later? The hypothesis is valid but others confirming this through experimentation not involving influences such as collapse of PMF may confirm the model.
· Lines 286 onwards: refers to data in Fig. 3 for Pbp2a. Figs. 1 and 2 show this PBP is dispersed and shows some concentration at the poles. The authors correctly state that the four stress conditions altered the probable location as shown in Fig. 3, noting that the control (3A) is less convincing that the data in Figs. 1 and 2, as explained by the different methods used. Please review the comments, as the ‘heat maps’ show NaCl stress increases the ‘density’ mid-cell, sorbitol also has has a similar impact but not as dramatic, vancomycin appears to concentrate but not at eht centre or poles while penicillin caused strong concentration mis-cell. Ref. 41 used B. subtilis 168 and included vancomycin and penicillin treatments in addition to nisin, showing no change in distribution of fluorescence which remained largely mid-cell and along the periphery to a lesser extent. The authors may wish to comment on this (if not already in the Discussion). The interesting data in Fig. 3 also showed vancomycin significantly lowered the proportion of the static proteins, which may support the contention that substrate availability is involved in vancomycin stress, given the mode of action. How reliable are the data?
3.4 deals with Pbp3
· Line 383: Fig. 4 A-C appear to have a similar location distribution, similar to seen for the earlier figures: disbursed, poles and some mid-cell foci (notwithstanding the differences in density in the ‘heat maps’. Both vancomycin and penicillin treatments obliterate or significantly reduce detection (respectively). How does this support the substrate availability model, given vancomycin decreases availability of substrate by binding while penicillin inhibits enzyme activity?
The authors need to look at their comments as the data interpretation appears to be biased towards supporting a substrate availability model: what would lead to an inability to detect Pbp3 after stress exposure? No substrate may account for this, but why are the locations of this protein not detected in 4D and E, and how is the data generated for vancomycin and penicillin stress in 4F? In the next two sections, the interpretation of the data includes lack of substrate availability is not an issue, for proteins located at the poles.
It would be extremely helpful to have a summary of the key data in terms of location changes and proportion of static/mobile data. Fig. 7 is very useful here, despite the caveat concerning the half-life of YFP. This would be very useful in the Discussion, as the Discussion generalises trends that are not substantiated from the data presented in Figs 3-7. Of particular note is reference to change in location of PBPs 3 and 4a, when the heat maps showed no likely location following vancomycin treatment and reduced probability for penicillin G treatment. Availability of substrate and its location during stress may well explain the observed changes in location and mobility. However, stress can also alter expression of a suite of proteins involved in stress responses, some core for cell wall integrity (turnover, repair) or the modulation of activity through interaction with the cell wall/PG machinery. As all of the PBPs were expressed at low level (from an introduced xylose promotor), their level of expression would have been static while the proportion of other components and their accessory proteins may well have altered. The Discussion concentrates on substrate availability as the working model (valid in part) but fails to explore alternative explanations, even when ref. [41] (albeit using strain 168 and different microscopy) showed no impact on location of Pbp2a with vancomycin or penicillin treatments. The authors are encouraged to explore some of these ideas in light of a summary table, and reduce generalisations that do not apply to all four of the PBPs.
Overall
Despite the above points, there is some very useful information in the submission, as it is using a single molecule tracking approach to explore the orchestrated movement of proteins following stress exposure. Once the interpretation of the data and avoidance of summation in light of one model is addressed, the manuscript should be reconsidered.

Reviewer 3 Report
Comments and Suggestions for Authors
We appreciate the authors' efforts in creating this important manuscript;
The study offers important new information about the location and dynamics of PBPs in Bacillus subtilis, especially in the presence of antibiotics and under stressful circumstances.
The study is well designed, methods are described in details, resulted are clearly presented, discussion fully interpreted findings, and conclusion successfully summarized the major findings. But I have some minor comments:
1- The abstract could be improved by emphasizing how the study's findings advance our knowledge of the mechanisms behind the synthesis and resilience of bacterial cell walls.
2- Conclusion: It is better to omit references and to offer suggestions that should be taken into account in subsequent research.
3- Reference No. 5 has been repeatedly cited. The discussion can be enriched by adding other references in the same context.
4- Typo errors:
-Line 32: “The cell not” correct to “the cell wall not”
- Throughout the text and reference list, bacterial species names must be
written in italic type phase “the attached file could be helpful”

Round 2
Reviewer 1 Report
Comments and Suggestions for Authors
Thanks for improving the manuscript with the recommendations.
There are still a fee typing errors.
Reviewer 2 Report
Comments and Suggestions for Authors
It was most useful for the authors to provide the comments to all 3 reviewers, although this is a little unusual as these comments are normally not seen by all reviewers before a manuscript is accepted. However, it did help to clarify some of my questions about inconsistencies in data interpretation in the first version of this submission, due to incorrect labelling in figures.
In version 2 of this submission, the authors have:
- Addressed some of the typographical/format issues raised before, but further revision is still needed for consistency in italics for Latin names, consistent use of italics for mVenus (when used as a gene versus protein), abbreviation of bacterial genus names after first instance of spelling out in full, not captialising trivial bacterial names (mycobacteria, not Mycobacteria when the genus is Mycobacterium), etc. Inconsistencies start from line 103.
- Provided clarity in phrasing at most of the points raised.
- Altered the Abstract in view of reviewer comments.
- Provided some further clarification to questions raised in the response to reviewer comments, but may not have altered the text of the manuscript to help future readers. Examples are:
- I asked about the vector used, given the reference was for a gfp vector whereas the current vector has mVenus. This was raised so that the MW of the bands in the Western blot could be interpreted. Providing the information that gfp and mVenus proteins are roughly the same size (approximately 27 kDa) so the predicted size of the fusion proteins in the supplementary data (81 to around 100 kDa, latter similar to PBP1A and 1B) should be clarified in the text somewhere.
- The unexplained band 40-50 kDa in the Western blot is a mystery but it is obvious that this would raise questions by readers, given the failure to verify expression of 2 of the 4 PBPs using specific antibodies for gfp. Some comment would be helpful, given the poor quality of the blot. An empty vector control was not included?
- References have been added to clarify methods but see below for further issues raised. I have not checked the appropriateness of every change.
Thank you for addressing these points. Further matters that require consideration follow.
- Line 183, legend of Figure 1. This now has references 36 and 37 for the structure of the 4 PBP from B. subtilis. 36 is a review, main information is from E. coli studies and this paper contains cartoons of models for E. coli PG turnover machinery. There is no source information for the structure of the PBP in B. subtilis. Ref. 37 is a study on alkaline shock of B. subtilis and showed (albeit not without questions on the experimental protocols used) that different PBP are expressed under stress and in a temporal fashion. Neither reference provides source information for Fig. 1A cartoon.
It is broadly accepted that the location of the PG machinery includes complexes of proteins linked to transmembrane components, protein in the cytoplasm and ones whose ‘job’ is essentially extracellular, in the PG matrix, but where the proteins are linked to the main PGEM. This includes the PBPs, which function outside the cell membrane but are in complexes (see Ref. 36 cartoons) and sometimes with transmembrane domains. The authors refer to PBP4a regarding controversy over whether this protein has a membrane ‘hook’, which is validly considered. However, a quick check in SignalP(V6) and DeepTMHMM, and UniProt (which may not show accurate data for signal and TM motifs) showed:
BSU_25000 (pbpA/=2A): no signal peptide, AA 1-23 inside, 24-48 TM helix, remained outside;
BSU_04140 (pbpC/=3): AA 1-20 SS, no TM helix predicted; lipoprotein signal peptide Sec/SPII; this means that the protein would be lipid-linked;
BSU_31490 (pbpD/=4): AA 1-6 inside, 7-28 TM helix, SS (UniProt) but SigP shows weak (score 0.4) Sec/SPI;
BSU_18350 (dacA/=4a): AA 1-19 signal peptide, Sec/SpI (strong score), no transmembrane regions predicted.
The authors need to consider the above as the analysis throws doubt on the validity of Fig. 1A and also raises questions about how the fusion proteins for PBP3 and 4a are processed to the normal location of the original proteins and whether insertion of mVenus would make these proteins (a) undetectable in whole cell Western blots or (b) functional. The databases predicting structures and locations of protein can be in error and are updated from time to time (most recent versions therefore used above) and the physical location needs to be backed by experimental data in addition to predicting structure/locations.
The authors need to provide accurate sources for any proposed cartoon, revise Fig. 1A and if Fig. 1 is revised, use the similarly-scaled images of cells for 1B currently in supplementary material (Fig. S4), as this allows direct comparison of cell shape/size (as noted by reviewer 1).
- In response to reviewer 1, the authors provided an image of a Bocillin assay, which detects binding of a fluorescent-tagged penicillin analogue which mimics the substrate. It was very useful to have this assay provided but it raises more questions. The control,
PY79 land 1, shows at least two bands which presumable correspond to PBPs 1A and B. There are 2 bands in lane 3, providing no evidence that a fusion protein has been expressed and consistent with Western blot data. White * in lanes 2 and 4 show poorly defied bands are the * is meant to indicate the presence of a fusion protein, but the images are not sharp or convincing. However, disappearance of the native size protein in the mVPBP4 does support expression of an active fusion protein, again consistent with the Western blot data for this fusion protein. However, the red asterisk (notating native protein band presence/absence) is also not convincing for PBP2a, as the band that is reduced is labelled as PBP2B while the band above this, PBP2a arrowed, is seen as still expressed. This new diagram raises concerns about what is expressed and active. Notwithstanding this, is the following accurate:
2a (has TM helix): mVenus fusion expressed (Western blot, but confusion in labels on Bocillin-FL new image), functional as it complements the deletion of paralog pbpH.
3 (has signal peptide, no TM helix): mV-fusion not expressed (Western blot, Bocillin-Fl assay), activity not confirmed as no paralog available to test complementation.
4 (has TM helix): mV-fusion expressed and functional (Western blot and Biocillin-FL)
4a (has Signal peptide, no transmembrane regions, optimal pH from literature pH 12): mV-fusion not expressed (Western blot), no data from Bocilin-FL, no complementation data.
However, Fig. 1B shows location of fusion proteins in cells. What is being detected? If not expressed and no evidence of functionality, would the fusion proteins for mV-3 and mV-4a still be able to bind to substrate?
I previously raised questions about whether the N-terminal insertion of mVenus would cause issues with these proteins, acknowledging that the authors also raised concerns for 4a. To be less obtuse, would proteins with signal peptides still be processed appropriately? Possibly, but needs to be considered in context of the points above. Would including a 27 kDa marker protein with the target protein alter the ability of the target protein to interact with its substrate, remain biologically active and change conformation, if required, to assume an active configuration or effective binding to other parts of the PG synthesis machinery so the machinery becomes ineffective? Some amidases are inactive, for example, unless they change configuration when binding to cell membrane complexes or need a companion protein for function. The authors need to consider these questions, given that they have not shown in their data that 2 of the 4 PBPs are functional or indeed expressed. If they cannot bind their substrate, is substrate binding the cause of the observed differences in the presence of penicillin and vancomycin, or is another part of the PG synthesis machinery being impacted as a primary response?
Can the authors consider the above in light of the lack of TM regions in the structure of the proteins.
Round 3
Reviewer 2 Report
Comments and Suggestions for Authors
The authors have responded to several of the points raised in the revised manuscript and have noted point of dispute regarding other matters. There are sufficient caveats in place and a vigorous discussion was most enjoyable.
Comments on the Quality of English LanguageNo issues